# A Novel Double Mosaic Virus-like Particle-Based Vaccine against SARS-CoV-2 Incorporates Both Receptor Binding Motif (RBM) and Fusion Domain

**DOI:** 10.3390/vaccines9111287

**Published:** 2021-11-05

**Authors:** Xinyue Chang, Andris Zeltins, Mona O. Mohsen, Zahra Gharailoo, Lisha Zha, Xuelan Liu, Senta Walton, Monique Vogel, Martin F. Bachmann

**Affiliations:** 1Department of Rheumatology and Immunology, University Hospital Bern, 3010 Bern, Switzerland; xinyue.chang@dbmr.unibe.ch (X.C.); mona.mohsen@dbmr.unibe.ch (M.O.M.); zahra.gharailoo@dbmr.unibe.ch (Z.G.); xuelan.liu@dbmr.unibe.ch (X.L.); monique.vogel@dbmr.unibe.ch (M.V.); 2Department of BioMedical Research, University of Bern, 3012 Bern, Switzerland; 3Latvian Biomedical Research & Study Center, Ratsupites 1, LV1067 Riga, Latvia; anze@biomed.lu.lv; 4Saiba GmbH, 8808 Pfäffikon, Switzerland; senta.walton@usz.ch; 5International Immunology Centre, Anhui Agricultural University, Hefei 230036, China; zhalisha@ahau.edu.cn; 6Jenner Institute, University of Oxford, Oxford OX3 7BN, UK

**Keywords:** COVID19, SARS-CoV-2, vaccine, virus-like particle, CuMV_TT_-DF

## Abstract

COVID-19 has emerged, and has rapidly become a major health problem worldwide, causing millions of mortalities. Vaccination against COVID-19 is the most efficient way to stop the pandemic. The goal of vaccines is to induce neutralizing antibodies against SARS-CoV-2 virus. Here, we present a novel double mosaic virus-like particle (VLP) displaying two independent neutralizing epitopes, namely the receptor binding motif (RBM) located in S1 and the fusion peptide (AA 817–855) located in S2. CuMV_TT_ virus-like particles were used as VLP scaffold and both domains were genetically fused in the middle of CuMV_TT_ subunits, which co-assembled into double mosaic particles (CuMV_TT_-DF). A single fusion mosaic particle (CuMV_TT_-FP) containing the fusion peptide only was used for comparison. The vaccines were produced in *E. coli*, and electron microscopy and dynamic light scattering confirmed their integrity and homogeneity. In addition, the CuMV_TT_-DF vaccine was well recognized by ACE2 receptor, indicating that the RBM was in native conformation. Both CuMV_TT_-FP and CuMV_TT_-DF vaccines induced high levels of high avidity IgG antibodies as well as IgA recognizing spike and RBD in the case of CuMV_TT_-DF. Both vaccine candidates induced virus-neutralizing antibodies indicating that the fusion peptide can independently induce virus-neutralizing antibodies. In contrast, CuMV_TT_-DF containing both RBM and fusion peptide induced a higher level of neutralizing antibodies suggesting that the new double mosaic vaccine candidate CuMV_TT_-DF consisting of two antigens in one VLP maybe an attractive candidate for scale-up in a bacterial fermentation process for clinical development.

## 1. Introduction

Severe acute respiratory syndrome coronavirus 2 (SARS-CoV-2), the cause of the COVID-19 pandemic, was first reported in Wuhan, China [1]. Meanwhile, the virus has infected over 200 million people worldwide and caused more than 4 million fatalities [2]. Despite extensive attempts to treat the disease, there is no curable medication available [3,4]. Therefore, global vaccination programs that aim to provide herd immunity are an ideal long-term solution for control of the pandemic.

SARS-CoV-2 utilizes the receptor binding domain (RBD) on spike protein to invade host cells. Like the SARS-CoV-1 virus, RBD of SARS-CoV-2 binds to the angiotensin-converting enzyme 2 (ACE2) receptor and is proteolytically cleaved to allow cellular entry of the virus [5]. This offers at least two points of attack for neutralizing antibodies: blocking the interaction of the receptor binding motif (RBM) within RBD with ACE2 [6], or blocking the proteolytic cleavage. There are currently more than 300 vaccine candidates in clinical and preclinical development [7], almost all of which target the spike protein [8,9] except for attenuated or inactivated whole virus [10,11,12,13]. There are several vaccines licensed for emergency usage, which are mostly based on nanoparticle-formulated mRNA, inactivated whole virus, or Adenovirus vectors [14].

While all these vaccine platforms have their clear advantages, they also have technical limitations. The main issue for mRNA vaccine is shortage of supply and the very low temperature required for long-term storage [15]. Vaccines based on inactivated whole virus are produced by conventional techniques. However, in the case of SARS-CoV-2, they induce rather low neutralizing responses, an observation compatible with the fact that most individuals infected with virus mount relatively low antibody responses despite sometimes rather high viral load. The same argument extends to attenuated viruses [16,17,18]. Adenoviruses-based vaccines targeting COVID-19 can easily be produced at large scale and can also be stored at conventional temperature (4 °C). However, it is difficult to apply more than one vaccine injection, which renders it challenging to maintain long-term antibody responses. In an interesting new development, recent studies have demonstrated that “mix and match” vaccines (ChAdOx1 nCoV-19 and BNT162b2) could elicit strong immune responses [19], which suggests that combining different vaccines may overcome drawbacks of individual vaccines. Therefore, more vaccine candidates would increase the chance to meet the demand of global vaccination.

Generally, neutralizing antibodies serve a predominant role in defending viral infections, in particular after vaccination [20,21]. In fact, studies have demonstrated that neutralizing antibodies protect animals and humans from SARS-CoV-2 infection [20,22,23]. Recently, clinical trials using neutralizing IgG antibodies such as cocktail REGN-COV2 [24] and LY-CoV555 [25] have shown positive results in treating COVID-19. In addition to IgG antibodies, IgA antibodies may dominate the mucosal defense in the respiratory tract [26]. Indeed, IgA has been shown to be important for neutralizing SARS-CoV-2 in infected patients [27,28,29]. Therefore, a good vaccine candidate should elicit strong neutralizing IgG as well as IgA antibody responses.

Interestingly, a study aiming to identify IgG immunodominant epitopes in spike protein claimed that two linear epitopes (AA 562–579 and AA 818–830) were recognized by convalescent patient sera [30]. Epitope AA 562–579 sits in RBD, but outside of the RBM region and represents a classical neutralizing epitope. Surprisingly, the second epitope AA 818–830 peptide is located in S2 protein and constitutes an important part of the previously described fusion peptides (AA 816–833) [5,31]. We therefore reasoned that it might be possible to generate a more effective vaccine candidate if such an epitope was to be included in addition to the classical RBD/RBM epitope.

CuMV_TT_ VLPs are optimized to activate T helper cells by incorporating a universal tetanus toxin (TT) epitope recognized by memory Th cells of almost every individual due to universal recognition as well as prior vaccination against TT. This epitope has been genetically fused internally to CuMV (cucumber mosaic virus) particles, which has created a powerful vaccine platform currently used for development of prophylactic and therapeutic vaccines in companion animals and humans [32,33,34,35]. Since the VLPs are produced by *E. coli* fermentation, one major advantage of vaccines based on CuMV_TT_ is their cost-effectiveness and scalability. Furthermore, CuMV_TT_ packages *E. coli* derived RNA, which represents the best stimulator of TLR7/8 in B cells [36].

To generate a vaccine candidate that effectively protects against SARS-CoV-2 virus, we produced a double mosaic particle by genetically fusing into individual CuMV_TT_ subunits the fusion peptide (AA 817–855) and RBM. This double mosaic particle represents a next generation version of the recently reported CuMV_TT_-RBD and CuMV_TT_-RBM vaccine candidates, which already demonstrated potent immunogenicity in vaccinated mice [37,38]. In contrast to these previously described VLP-based vaccines, the new CuMV_TT_-DF vaccine candidate targets the RBD-ACE2 interaction interface as well as epitope required for fusion of the virus with the endosomal membrane. The CuMV_TT_-DF immunized mice showed strong IgG and IgA antibody responses, which could effectively neutralize SARS-CoV-2 virus. To our knowledge, this is the first time two different epitopes have been displayed on a single VLP by genetic fusion, extending the potential of VLPs as a better vaccine platform.

## 2. Materials and Methods

### 2.1. Ethics Statements

All studies involving animal were subject to prior approval by the respective local ethics committees. For studies using mice, methods were performed in accordance with regulations and guidelines of the Cantonal Veterinary Office Bern, Switzerland. Methods were approved by the animal ethics research committee of the Cantonal Veterinary Office Bern, Switzerland as part of the standard operating procedure for study approval for project (license BE70/18). No human material was used in this study.

### 2.2. Vaccine Production and Characterization

*E. coli* ER2566 containing pETDu-CMV3d-nCoV-FP-CMVtt or pET28-CMVB3d-nCoV-M-CMV-FP were used to produce CuMV_TT_-FP and CuMV_TT_-DF vaccines, respectively. Bacteria were cultured in 2TY medium (1.6% tryptone, 1% yeast extract, 0.5% NaCl) with 100 mg/L Ampicillin at 30 °C until OD_600_ = 0.8. Then 0.2 mM IPTG (Isopropyl β-D-1-thiogalactopyranoside) and 5 mM MgCl_2_ were added to induce protein expression at 20 °C. Biomass was harvested by centrifugation 18 h after induction, and suspended in lysis buffer (20 mM Tris, 5 mM EDTA, 5 mM Et-SH, 5% glycerol, 10% sucrose, pH 8.0). VLP vaccines were purified as described previously with minor adjustment [32]. Briefly, bacteria were disrupted with sonication and the lysate was then rotated at 10 rpm, 4 °C overnight. Afterwards, VLP-containing supernatants were separated from cell debris with 10,000 rpm centrifugation for 10 min.

VLPs were purified by applying the supernatant to sucrose gradients (20–60% sucrose in buffer: 20 mM Tris, 2 mM EDTA, 5% glycerol, 0.5% Triton X100) and then centrifuged for 6 h at 25,500 rpm, 18 °C (Beckman SW32). Afterwards, fractions were collected and analyzed on SDS-PAGE gel, and VLP-containing fractions were 1:1 diluted in buffer (20 mM Tris-HCl, 2mM EDTA, 5% glycerol) and then sedimented using 50,000 rpm centrifugation for 4 h at 4 °C. Finally, VLPs were obtained by dissolving pellets in buffer (20 mM Tris-HCl, 5 mM EDTA, 5% glycerol). The quality and purity of VLPs were examined by means of SDS-PAGE gel analysis, agarose gel analysis, dynamic light scattering (DLS) and transmission electron microscope (TEM).

The average size of particles was determined by DLS. Briefly, VLPs were diluted to 1 mg/mL and analyzed by a Zetasizer Nano ZS instrument (Malvern Instruments Ltd., Malvern, UK) [32]. Three repeats were performed.

VLP samples were negatively stained before TEM observation. Firstly, 5 ul of each suspension were adsorbed on glow discharged and carbon coated 400 mesh copper grids (Plano, Wetzlar; Germany) for 1 min. After washing 3 times by dipping in H_2_O, grids were stained with 2% uranyl acetate solution (Electron Microscopy Sciences, Hatfield, PA, USA) for 45 s. The excess fluid was removed by gently pushing them sideways to filter paper. Stained samples were then examined with a transmission electron microscope (Tecnai Spirit, FEI, Hillsboro, OR, USA) at 80 kV and equipped with a digital camera (Veleta, Olympus, Münster, Germany).

### 2.3. Mice Immunization

*BALB/claHsd* (Envigo, Horst, The Netherlands) mice were purchased and kept in SPF animal facility (Department of Biomedical Research, University of Bern, Bern, Switzerland) according to Cantonal Veterinary guidelines of Bern. Female mice (8–12 weeks) were immunized subcutaneously with 100 ug purified CuMV_TT_-FP, CuMV_TT_-DF vaccine and boosted with same dose at 21 days after priming. Serum samples were collected every week until d49. Five mice were used per group.

### 2.4. Recombinant Receptor Binding Domain (RBD) of Spike Protein Production

Gene encoding the RBD protein (319R-541F, 223 AA) with a His-tag in C-terminus was synthesized by Twist Bioscience HQ (South San Francisco, CA, USA) and cloned into eukaryotic expression plasmid pTwist CMV BetaGlobin WPRE Neo. The expression plasmid was amplified in XL-1 Blue electro-competent cells and then extracted with PureLink™ HiPure Plasmid Filter Maxiprep Kit (Invitrogen, catalog K210016, Waltham, MA, USA) in accordance with product manual. The Expi293F cells (50 mL, 3 × 10^6^ cells/mL) were transfected with 50 ug DNA using transit ExpiFectamine 293 transfection kit (Gibco, catalog A14524, Amarillo, TX, USA). Cell culture supernatants containing RBD protein were collected by 4000 rpm, 20 min centrifugation. RBD protein was purified from the supernatants by passing through His-Trap HP column (GE Healthcare, catalog 17-5248-01, Marlborough, MA, USA) after 0.22 um filtering. Then the purified RBD protein was analyzed in SDS-PAGE gel and stored at −20 °C.

### 2.5. ELISA and Avidity ELISA

ELISA was performed to determine the immunogenicity of vaccines. Corning half area 96-well plates were coated with 1 ug/mL recombinant RBD protein, spike protein (Sino Biological, catalog 40589-V08H4) or spike S2 protein (Sino Biological, catalog 40590-V08B) overnight at 4 °C. Then plates were blocked with PBS-0.15% Casein for 2 h at room temperature, after which mice sera were added and 1:3 serial diluted starting from 1:20 dilution in PBS-0.15% Casein. After incubation at room temperature for one hour, goat anti-mouse IgG-POX antibody (Jackson Immunoresearch, catalog 115-035-071) was added and incubated for one hour. Finally, plates were developed with tetramethylbenzidine (TMB) in citrate buffer, stopped with 1 M H_2_SO_4_ solution and read at OD_450nm_.

Same procedures were applied to determine spike-specific IgA antibodies, with exception of goat anti-mouse IgA-POX antibody (ICN Cappel, catalog 55549) as secondary antibody. To eliminate the interference of IgG antibodies to IgA determination, IgG antibodies were removed from serum samples by incubating serum with Protein G magnetic beads (Thermo Scientific, catalog 88847) at room temperature for 10 min.

To determine if the vaccines were able to be recognized by angiotensin-converting enzyme 2 (ACE2) receptor, the plate was coated with recombinant ACE2-His tag (Sino Biological, catalog 10108-H08H). Then vaccines were added and diluted after blocking the plate with PBS-0.15% Casein. Afterwards, mouse anti-CuMV_TT_ IgG (home-made from hybridomas) was used to detect the bound vaccine, followed by adding sheep anti-mouse IgG-POX antibody (ICN Cappel, catalog 55558). Finally, the plate was developed and stopped as described above.

To analyze the binding of monoclonal anti-RBD antibody to RBM peptides in CuMV_TT_-DF, half-well Corning plate was coated with anti-CuMV_TT_ antibody overnight at 4 °C. Then plate was blocked with PBS-0.15% Casein for 2 h at room temperature, after which CuMV_TT_-DF (CuMV_TT_ as a control) vaccine was added to plate and serially diluted. After incubation at room temperature for an hour, human anti-RBD antibody (Sanyou biopharmaceuticals, catalog AHA004, Shanghai, China) was added and incubated for one hour. Finally, goat anti-human IgG conjugated with POX (Nordic MUbio, catalog GAHu/IgG(Fc)/PO, Susteren, The Netherlands) was used as detection antibody. After an hour incubation, plate was developed with TMB and stopped with 1 M H_2_SO_4_.

To test the affinity of specific antibodies in immunized mouse serum, avidity ELISA was performed. To this end, two parallel plates were performed as above, except one washing step before adding secondary antibody. One plate was washed with PBS-0.05% Tween-7 M urea 3 times, while the other with PBS-0.05% Tween only, in between which plates were washed with PBS-0.01% Tween. The weakly bound antibodies are washed away with 7 M urea. Avidity index was calculated as ratio of OD (PBS-0.05% Tween-7 M urea) to OD (PBS-0.05% Tween) with same dilution [39].

### 2.6. SARS-CoV-2 Virus Neutralization Assay

Viral neutralization assays were performed as described previously [37]. In brief, serum samples were firstly heat-inactivated at 56 °C for 30 min. Subsequently, the samples were diluted 2-fold starting from 1:20 dilution until 1:160. 100 TCID50 of SARS-CoV-2/ABS/NL20 was added to each diluted serum and incubated for 1 h at 37 °C. Afterwards, the mixtures were added on a monolayer of Vero cells and incubated for additional 4 days at 37 °C. Finally, the wells were inspected for presence of cytopathic effect (CPE). Titer was expressed as the highest dilution of the serum that fully inhibits the formation of CPE.

## 3. Results

### 3.1. Purified CuMV_TT_-FP and CuMV_TT_-DF Vaccines Are of Homogenous Composition, Packaged with RNA, and in Correct Conformation

The engineered CuMV_TT_-FP and CuMV_TT_-DF vaccines were designed to be mosaic particles, which consist of two different subunits as illustrated in Figure 1A: mosaic particle consisting of wild type CuMV_TT_ and CuMV_TT_ fused with fusion peptide (AA 817–855) (CuMV_TT_-FP); or double mosaic particle consisting of CuMV_TT_ fused with RBM (AA 437–508) and CuMV_TT_ fused with the fusion peptide (AA 817–855) (CuMV_TT_-DF). This latter mosaic particle contains no wild-type subunit. Briefly, we could yield 2.7 mg CuMV_TT_-DF VLPs per gram biomass. We have shown previously for CuMV_TT_-RBM that yields were approximately 50-fold higher in bioreactors [38]. As shown in Figure 1B, SDS-PAGE gel demonstrated the correct sizes of the purified subunits of CuMV_TT_-FP (25 kD and 31.4 kD) and CuMV_TT_-DF (31.4 kD and 35.3 kD) vaccines. According to densitometric analysis, we conclude that RBM epitope accounted for 57% subunits and FP epitope for 43% in CuMV_TT_-DF vaccine, indicating that there are 103 RBM and 77 FP epitopes in each particle approximately. In addition, the self-assembled VLPs spontaneously pack negatively-charged prokaryotic ssRNA which stabilizes the viral structure. Accordingly, Figure 1C shows in agarose gel analysis that both CuMV_TT_-FP and CuMV_TT_-DF vaccines incorporate RNA, resulting in stable VLP vaccines and capacity to stimulate innate pattern receptors TLR7/8 [36]. The long-term stability of CuMV_TT_-DF vaccine was confirmed by SDS-PAGE and agarose gel after storing the vaccine at 4 °C for 6 months (Appendix A). In addition, we analyzed the CuMV_TT_-DF vaccine after 12 months storage at 4 °C. TEM image in Appendix A illustrates that CuMV_TT_-DF shows intact spherical viral structure identical to newly produced VLPs. Similarly, DLS also demonstrates homogeneous particles with an apparent diameter of 70 nm. These results indicate that CuMV_TT_-DF vaccine is stable at 4 °C for at least one year.

Furthermore, the purified vaccines were found to be homogenous by DLS and the majority VLPs were approximately 70 nm in diameter (Figure 1D). Similarly, transmission electron microscopy (TEM) analysis illustrated the vaccines were in typical spherical virus shape (Figure 1E). Taken together, the *E. coli* produced CuMV_TT_-FP and CuMV_TT_-DF vaccines successfully formed intact and stable virus-like particles that incorporated RNA.

To further characterize the vaccine candidates, the ability to bind to the viral receptor ACE2 was examined by ELISA. The RBM-containing CuMV_TT_-DF vaccine was recognized and bound by ACE2 (Figure 1F), suggesting the RBM was in natural conformation. As expected, CuMV_TT_-FP vaccine did not show binding to ACE2. Similarly, monoclonal anti-RBD antibody could recognize CuMV_TT_-DF (Figure 1G), confirming the fact that RBM in CuMV_TT_-DF vaccine is in native conformation.

### 3.2. CuMV_TT_-FP and CuMV_TT_-DF Vaccines Demonstrated High Immunogenicity in Mice

To examine the immunogenicity of vaccines in vivo, *BALB/c* mice were immunized and serum samples were collected as shown in Figure 2A. Our previous work on another VLP vaccine CuMV_TT_-RBM demonstrated that 100 μg VLP vaccine elicits the best antibody responses [38]. Therefore, we performed our immunization with this dose as well. As expected, CuMV_TT_-FP failed to induce RBD-specific antibodies in immunized mice. In contrast, mice immunized with CuMV_TT_-DF generated antibodies against RBD, and the titers further increased after booster vaccination (Figure 2B). Consistently, both CuMV_TT_-FP and CuMV_TT_-DF vaccines induced spike S2 protein-specific IgG antibodies in mice and the titers increased with time, in particular after booster immunization (Figure 2C). Furthermore, strong spike protein-specific antibodies were induced by both CuMV_TT_-FP and CuMV_TT_-DF (Figure 2D). Titers induced by CuMV_TT_-DF at d49 were higher than those induced by CuMV_TT_-FP immunization, indicating the CuMV_TT_-DF induces overall better anti-viral responses. In summary, both CuMV_TT_-FP and CuMV_TT_-DF vaccines were able to induce potent IgG antibody responses against SARS-CoV-2.

High affinity antibodies are needed for good viral neutralization. To examine the avidity of IgG antibodies induced by CuMV_TT_-FP and CuMV_TT_-DF vaccines, avidity ELISA assays were performed. As shown in Figure 3, levels of high-avidity antibodies against RBD (Figure 3B), S2 protein (Figure 3D) and spike protein (Figure 3F) in sera from CuMV_TT_-DF immunized mice were elevated in particular after booster immunization. Similarly, avidity indexes against S2 protein (Figure 3C) and spike (Figure 3E) were significantly increased after boosting. Importantly, around 50% of IgG antibodies remain bound to spike protein after 7 M urea wash, indicating the vaccine-elicited IgG antibodies are of very high avidities. Therefore, in addition to high antibody titers, CuMV_TT_-FP and CuMV_TT_-DF vaccines induce antibodies of high avidity to SARS-CoV-2.

Finally, IgA antibodies which typically dominate local anti-viral infections in the respiratory tract were determined. As shown in Figure 4, IgA antibodies were high after booster vaccination and stayed at high levels at least until d49. Taken together, CuMV_TT_-FP and CuMV_TT_-DF vaccines are highly immunogenic and induce IgG antibodies of high avidity and IgA in mice.

### 3.3. CuMV_TT_-DF Vaccine Induced Neutralizing Antibodies in Mice

After confirming the potent immunogenicity of CuMV_TT_-FP and CuMV_TT_-DF vaccines, we next tested the efficiency of the antibodies to neutralize SARS-CoV-2 virus. Figure 5 shows that neutralization titers induced by both CuMV_TT_-FP and CuMV_TT_-DF were significantly higher than control sera (buffer and CuMV_TT_ immunized mice [38]), demonstrating that both CuMV_TT_-FP and CuMV_TT_-DF vaccines were able to induce neutralizing antibodies. In addition, titers of CuMV_TT_-DF immunized mice were higher than those from CuMV_TT_-FP immunized mice, suggesting that RBM is essential for a vaccine to induce high levels of neutralizing antibodies but that peptide AA 817–855 alone is also able to do so. We have shown previously that CuMV_TT_-RBM vaccine, which incorporates wild type CuMV_TT_ and CuMV_TT_-RBM subunit, elicited comparable neutralization levels, indicating that RBM is the major contributor to neutralization [38]. Nevertheless, the neutralizing anti-FP antibodies may help avoid viral escape from neutralizing antibodies recognizing RBM alone. In summary, CuMV_TT_-DF vaccine could induce antibodies that efficiently neutralize SARS-CoV-2, and thus be an alternative COVID19 vaccine candidate for further clinical development.

## 4. Discussion

Several vaccine candidates based on VLPs against SARS-CoV-2 virus have been tested by us, all of which aimed to induce neutralizing antibodies. The CuMV_TT_-RBD was our first successful vaccine candidate generated by chemical coupling of RBD protein in native conformation from eukaryotic expression to CuMV_TT_ particles, which induced high neutralizing antibodies in immunized mice [37], similarly to RBD displayed on AP205 VLPs by the Spy-Catcher method [40]. On this premise, we next genetically fused the RBM, which has no post-transcriptional modifications, into the middle of CuMV_TT_ subunit and produced the mosaic CuMV_TT_-RBM vaccine in *E. coli*. For this vaccine candidate, we could show that it can be produced at large scale from a stable *E. coli* fermentation process at low costs in short time [38]. Notably, the CuMV_TT_-RBM vaccine demonstrated as high an immunogenicity as CuMV_TT_-RBD, even several months post vaccination and induced high levels of neutralizing antibodies. In addition, another vaccine candidate based on AP205 VLPs C-terminally fused to RBM (AP205-RBM) demonstrated potent immunogenicity in mice and induced neutralizing antibodies [41].

Even though RBM/RBD are the main neutralizing epitopes on the spike protein, there are additional “minor” epitopes. Even though they are less potent epitopes, they are completely independent of RBM/RBD and therefore may add to more robust neutralization. We therefore combined the minor epitope AA 817–855 situated in S2 with RBM, located in S1 forming a “double fusion” CuMV_TT_-DF. Such a vaccine may elicit neutralizing antibodies that block spike-ACE2 interaction as well as blocking fusion of the virion with endosomal membrane. To assess whether AA 717–855 alone can induce neutralizing antibodies, CuMV_TT_-FP that incorporates this peptide alone was generated and shown to induce neutralizing antibodies, albeit at moderate levels. Thus, CuMV_TT_-DF induces neutralizing antibodies against two independent epitopes, both conferring neutralization of SARS-CoV-2, and likely reducing the likelihood of viral escape.

Recently, Cohen et al. reported the Spy/Catcher system to convey RBDs from SARS-CoV-2 and other Sarbecoviruses on the VLP scaffold, which induced cross-reactive immune responses in mice [42]. Here, instead, we present a direct technique by genetically fusing independent epitopes into one particle. This potential of displaying several antigens on a single VLP may be employed to a variety of novel vaccine candidates. For example, Cervarix and Gardasil vaccines may be optimized to be a “multi-fusion” vaccine [43], instead of mixing individual Human Papillomavirus type 16 and type 18 VLPs together.

Although the CuMV_TT_-FP vaccine candidate was able to induce antibodies against spike protein in mice, the mice sera neutralized SARS-CoV-2 at low levels in vitro, indicating the chosen epitope was indeed “minor” and not as potent as anti-RBM antibodies. In contrast, sera of mice immunized with CuMV_TT_-DF vaccine were able to completely neutralize SARS-CoV-2 virus, indicating that RBM was a key epitope to be included in vaccine candidates. The IgG antibodies elicited by CuMV_TT_-DF vaccine were of high avidity to the spike protein after the booster, and firmly bound to the spike protein, even after 7M urea wash. This might be an advantage over natural infection, after which the antibodies in blood of convalescent patients only bound RBD weakly to the RBD/spike [17,44] and exhibited much lower avidity to the spike protein than antibodies induced by RNA vaccination [18]. One plausible reason for the better quantity and quality after CuMV_TT_-DF vaccination than SARS-CoV-2 virus could be its structural features [45]. The long distance (25 nm) between spikes on the surface of SARS-CoV-2 virus is too large for optimal B cell activation; indeed, optimal antibody responses are induced by epitopes that are rigidly spaced by 5–10 nm, a distance that is found on almost all other viruses [46] and is recognized as pathogen-associated structure patterns (PASP) [47]. In contrast to the virus SARS-CoV-2, the CuMV_TT_-DF vaccine candidate here presents two epitopes in a repetitive and rigid pattern with an optimal distance of 5–10 nm, as described above to be optimal.

In addition to strong IgG antibodies, IgA was found in immunized mice sera and significantly enhanced after booster immunization. High titers of IgA antibodies in combination with IgG has been correlated to robust SARS-CoV-2 neutralization in infected humans [44]. The presence of serum IgA is, however, not always linked to mucosal IgA and it is necessary to establish whether our vaccine candidate induces mucosal IgA in humans.

## 5. Conclusions

Here we report for the first time a VLP-based vaccine candidate, which displays two different genetically introduced antigens on the surface. With this method, both the major neutralizing RBM epitope in S1 and the independent minor epitope AA817–855 on S2 protein were successfully incorporated in CuMV_TT_, resulting in CuMV_TT_-DF vaccine. The CuMV_TT_-DF vaccine was shown to be stable, homogenous, and packaged with RNA. In addition, it demonstrated strong immunogenicity and induced robust neutralizing antibodies in mice. Moreover, it may be expected that antibodies are less susceptible to viral escape as both epitopes are independent. In conclusion, CuMV_TT_-DF vaccine may not only represent an attractive candidate for a COVID19 vaccine, but also be a pioneer for VLPs displaying multiple antigens by genetic fusion.

## Figures and Tables

**Figure 1 vaccines-09-01287-f001:**
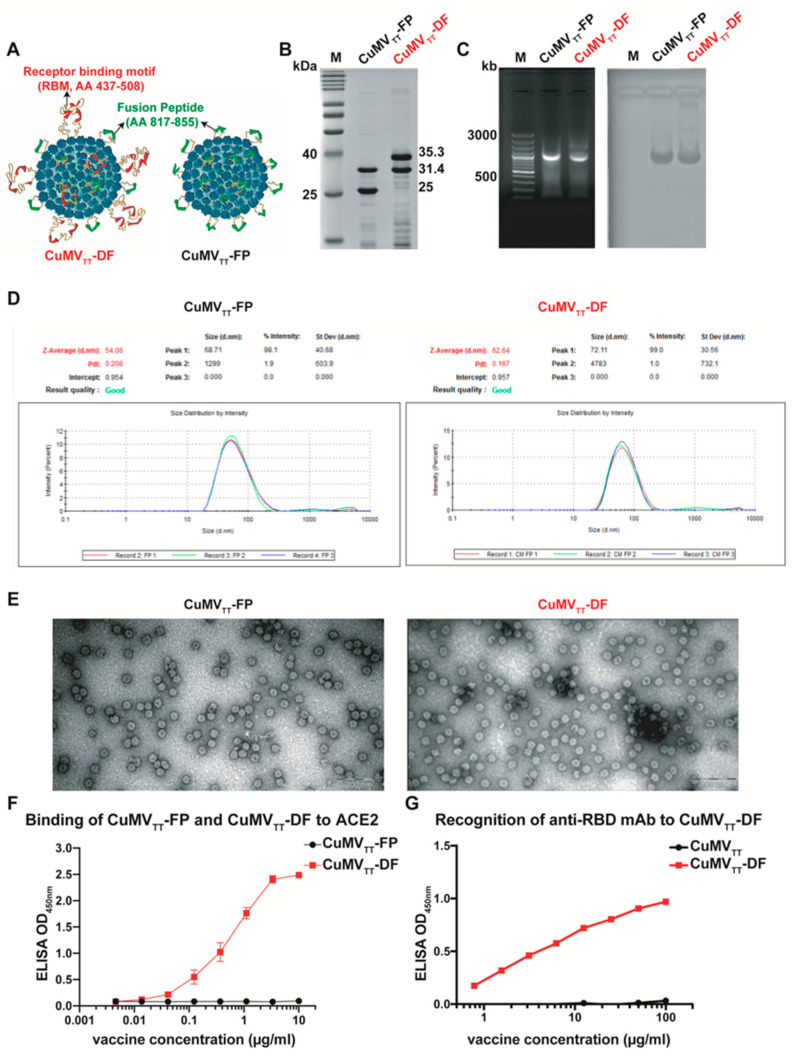
Characterization of CuMV_TT_-FP and CuMV_TT_-DF vaccines. (**A**): Design illustrations of CuMV_TT_-FP and CuMV_TT_-DF vaccines; (**B**): Analysis of purified CuMV_TT_-FP and CuMV_TT_-DF vaccine on SDS-PAGE gel; (**C**): Analysis of prokaryotic RNA incorporated in CuMV_TT_-FP and CuMV_TT_-DF VLPs on agarose gel, which was recorded under UV light (left) and afterwards, stained with Coomassie blue (right); (**D**): Determination of size distribution of CuMV_TT_-FP and CuMV_TT_-DF vaccine particles with dynamic light scattering; (**E**): Transmission electron microscopy analysis of CuMV_TT_-FP and CuMV_TT_-DF vaccines; (**F**): Binding of CuMV_TT_-FP and CuMV_TT_-DF vaccines to ACE2 receptor assessed by Sandwich ELISA; duplicates were performed. (**G**): Binding of anti-RBD monoclonal antibody to CuMV_TT_-DF vaccine assessed by Sandwich ELISA. OD_450nm_ are displayed for serially diluted vaccine solutions.

**Figure 2 vaccines-09-01287-f002:**
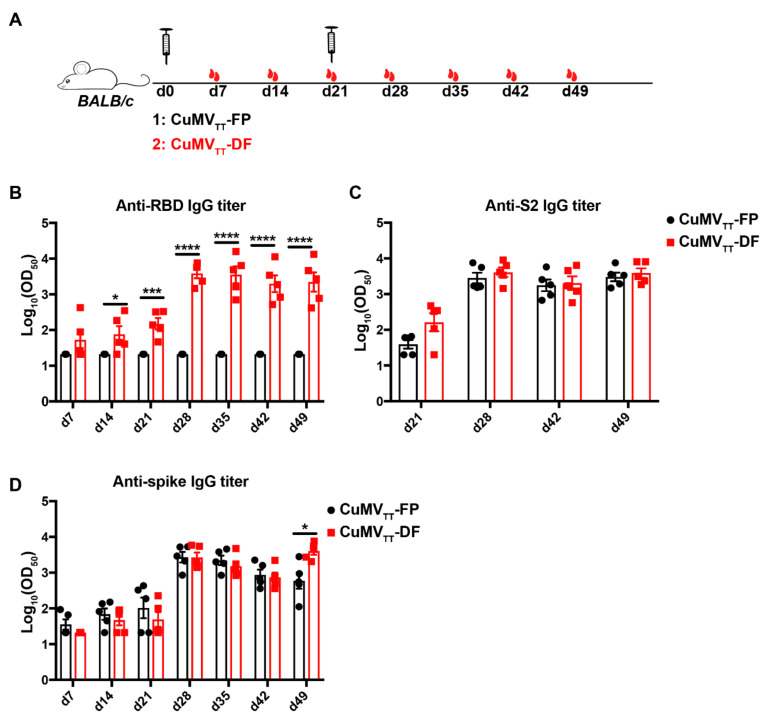
Immunogenicity examination of CuMV_TT_-FP and CuMV_TT_-DF vaccines in mice. (**A**): Immunization regimen: *BALB/c* mice were subcutaneously injected with 100 ug CuMV_TT_-FP or CuMV_TT_-DF vaccine at d0 and d21. Serum samples were collected every week until d49 (*n* = 5). (**B**–**D**): IgG antibody titers of immunized mice against RBD (**B**), S2 protein (**C**) and spike protein (**D**). Unpaired *t*-test analysis was performed in GraphPad Prism 7. α = 0.05 and statistical significance were displayed as *p* ≤ 0.05 (*), *p* ≤ 0.005 (***), *p* ≤ 0.001 (****).

**Figure 3 vaccines-09-01287-f003:**
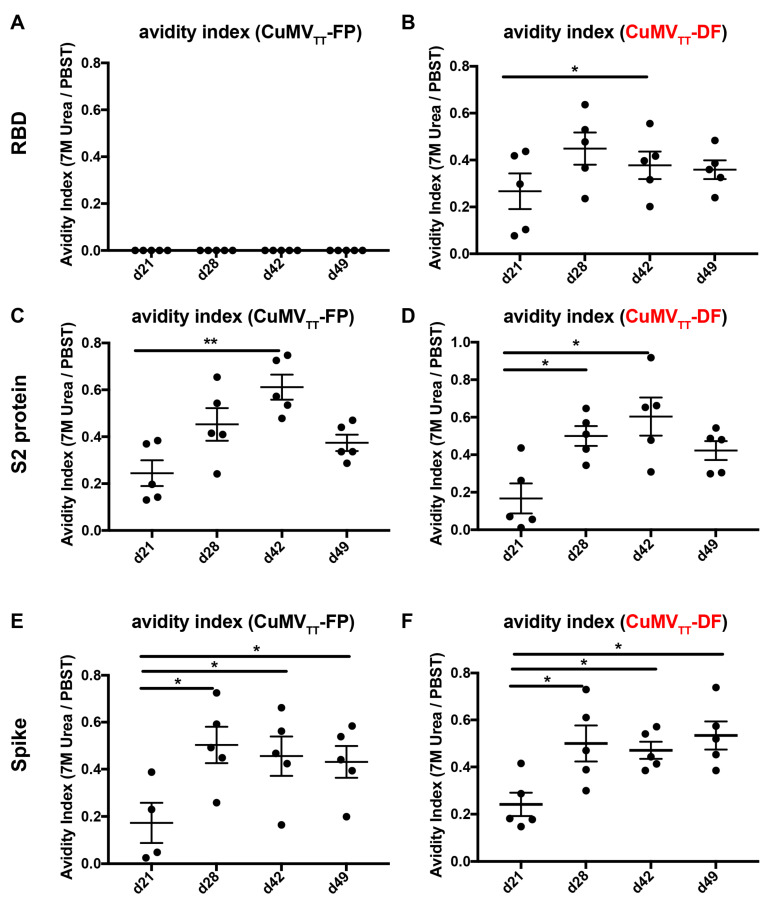
Avidity indexes of IgG antibodies in mice before and after booster immunization of CuMV_TT_-FP and CuMV_TT_-DF (right panel, indicated as red) measured by avidity ELISA (*n* = 5). (**A**,**B**): Avidity indexes of IgG antibodies for RBD from CuMV_TT_-FP (**A**) or CuMV_TT_-DF (**B**) immunized mice; (**C**,**D**): Avidity indexes for spike S2 protein of CuMV_TT_-FP (**C**) or CuMV_TT_-DF (**D**) elicited IgG antibodies; (**E**,**F**): Avidity indexes for spike S2 protein of CuMV_TT_-FP (**E**) or CuMV_TT_-DF (**F**) stimulated IgG antibodies. Paired t-test in GraphPad Prism 7 was used for statistical analysis. α = 0.05 and statistical significance were displayed as *p* ≤ 0.05 (*), *p* ≤ 0.01 (**).

**Figure 4 vaccines-09-01287-f004:**
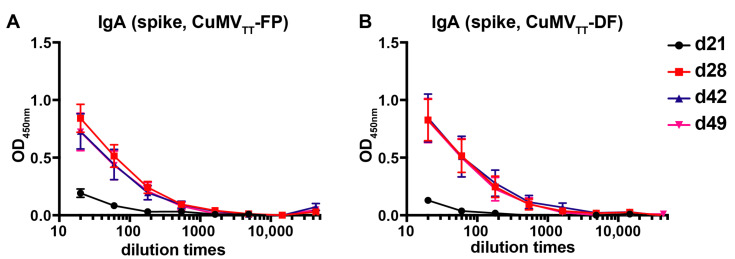
IgA antibodies in CuMV_TT_-FP (**A**) and CuMV_TT_-DF (**B**) immunized mice sera against spike protein (*n* = 5) before and after booster immunization. Shown are curves of ELISA OD_450nm_ values according to serum dilution, mean ± SEM.

**Figure 5 vaccines-09-01287-f005:**
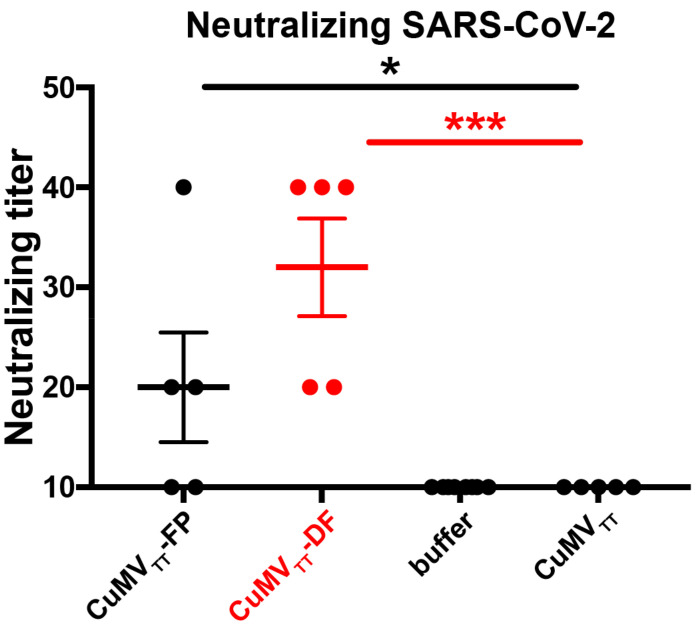
Titers of mice sera at d49 after CuMV_TT_-FP (*n* = 5), CuMV_TT_-DF (*n* = 5), buffer (*n* = 7) or CuMV_TT_ (*n* = 5) [38] immunization to neutralize SARS-CoV-2 virus. The titer was expressed as highest serum dilution times to inhibit 100% cytopathogenic effect (CPE) of cells infected with SARS-CoV-2 virus. Unpaired *t*-test analysis was performed in GraphPad Prism 7. α = 0.05 and statistical significance were displayed as *p* ≤ 0.05 (*), *p* ≤ 0.005 (***).

## Data Availability

All data generated or analyzed during this study are included in this published article (and its Appendix A).

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
