# Peer review of "A Novel Double Mosaic Virus-like Particle-Based Vaccine against SARS-CoV-2 Incorporates Both Receptor Binding Motif (RBM) and Fusion Domain"

_vaccines, 2021, doi:10.3390/vaccines9111287_

Round 1
Reviewer 1 Report
This manuscript describes the engineering and characterization of hybrid virus-like particles displaying two different domains/epitopes from the SARS-CoV-2 spike protein. A previous manuscript characterized VLPs displaying the Spike receptor binding motif (RBM) -- the main advance of this manuscript is the generation of hybrid VLPs that also display the fusion peptide. Overall, the manuscript is quite solid, but I do have a few comments/suggestions to improve the manuscript.
- Can the authors calculate the average number of copies of the FP and RBM epitopes per particle?
- Showing the VLPs displaying the RBM can bind to soluble ACE2 suggests that at least some of the displayed RBM is natively folded. However, there is a formal possibility that RBM exists in multiple conformations on the surface of the VLP. Is there a more quantitative approach for addressing whether the majority of RBM is in a native conformation? Perhaps this could also be addressed by measuring the binding of monoclonal antibodies that target this domain?
- One advantage of bacterial expression is the ability to produce vaccine at scales that are not possible in eukaryotic expression systems. It would be helpful to include some information on the yield of VLPs per liter of culture.
- The dose of vaccine utilized in these studies is quite high (100µg). Presumably this dose was selected based on data presented in the author's 2021 manuscript in Allergy. Nevertheless, additional discussion of dose would be beneficial to the manuscript.
- One control that is missing throughout the paper is mice immunized with wild-type CuMVtt VLPs. While I don't think that this is an essential control for many of the experiments, it would be helpful to include this control in the neutralization experiment reported in Figure 5. In addition, it would also be helpful to also show neutralization data from mice that received CuMVtt-RBM VLPs. (Since this data has been previously reported, it would be fine to just cite that data.)
Author Response
We thank the reviewer for all the comments, which help us to improve our manuscript. We have answered reviewer's comments point-to-point and corrected the manuscript accordingly. Please find attached.

Reviewer 2 Report
Ovidiu Radu
Assistant Editor
Dear Dr. Radu
Greetings,
Regarding reviewing of the article, entitled ”A novel double mosaic virus-like particle-based vaccine against SARS-CoV-2 incorporates both receptor binding motif (RBM) and fusion domain” which is submitted to the Journal Vaccines, I came to the following conclusion.
In my opinion, the article has a suitable novelty and can be published in this journal, provided that the following questions are answered correctly. My questions are as follows:
- As far as I know, RBD has two disulfide bonds, and expression in the E. coli system, whose cytoplasmic environment is reducing, prevents the formation of disulfide bonds. However, you have shown that the expression of your fusion protein in this prokaryotic system has led to its native conformation. Please explain why.
- Regarding the peptide fusion used in this paper, AA 817-855, is the rationale for using this peptide considered correct? Because, as you said, this peptide attaches and fuses the virus membrane to the endosomal membrane and releases the virus into the cytoplasm. So when the virus enters the endosome, can the antibody against this part enter the endosome along with the antigen? Is this design basically correct? I do not know if antibodies remain attached to the viral particle inside the endosome
- MgCl2 has been used to express VLP-based proteins other than IPTG. What is the reason for using this substance?
- What is the reason for using the following compounds in your Lysis buffer? :
Et-SH, 5% glycerol, 10% sucrose
- Why is a sucrose gradient used to purify VLPs? While your constructs have histidine tags and can be easily purified with commercial Nickel beads.
-- When you analyze each VLP-based vaccine, you will see 2 bands in your SDS-PAGE. Shouldn’t it be just one band as the VLP is covalently fused to antigens? The band 31KD is common in both (in figure 1B). Explain, please.
Reviewer 3 Report
The paper is an interesting contribution to the vaccine field. I would recommend some additional discussion on the following items:
Line 216, "The long-term stability of CuMVTT-DF vaccine was confirmed by SDS-PAGE and agarose gel after storing the vaccine at 4ËšC for 6 months". The long-term stability study presented is pretty naive. Further characterization using analytical SEC-HPLC, DLS, and TEM would be required or at least justified why was not performed. Also, longer incubation periods and accelerated degradation programs (25, 42°C) should be performed or justified otherwise.
There was no mention/discussion to the most important issue of the current SARS-CoV-2 pandemic i.e., how vaccines would deal with variants specially the Delta. How your multiple antigen vaccine strategy would fit in this scenario? Some discussion would improve the work.
Round 2
Reviewer 1 Report
The authors have addressed my concerns.